# Misreporting of Physical Activity and Sedentary Behavior in Parents-to-Be: A Validation Study across Sex

**DOI:** 10.3390/ijerph18094654

**Published:** 2021-04-27

**Authors:** Tom Deliens, Vickà Versele, Jasper Jehin, Eva D’Hondt, Yanni Verhavert, Peter Clarys, Roland Devlieger, Annick Bogaerts, Dirk Aerenhouts

**Affiliations:** 1Department of Movement and Sport Sciences, Faculty of Physical Education and Physiotherapy, Vrije Universiteit Brussel, Pleinlaan 2, 1050 Brussels, Belgium; Vicka.versele@vub.be (V.V.); jasperjehin@hotmail.com (J.J.); eva.dhondt@vub.be (E.D.); Yanni.verhavert@vub.be (Y.V.); peter.clarys@vub.be (P.C.); dirk.aerenhouts@vub.be (D.A.); 2Research Unit Woman and Child, Department of Development and Regeneration, KU Leuven, Oude Markt 13, 3000 Leuven, Belgium; roland.devlieger@uzleuven.be (R.D.); Annick.bogaerts@kuleuven.be (A.B.); 3Obstetrics and Gynaecology, University Hospitals KU Leuven, Herestraat 49, 3000 Leuven, Belgium; 4Centre for Research and Innovation in Care (CRIC), Faculty of Medicine and Health Sciences, University of Antwerp, Universiteitsplein 1, 2610 Wilrijk, Belgium; 5Faculty of Health, University of Plymouth, Devon PL4 8AA, UK

**Keywords:** validity, self-report, questionnaire, accelerometry, pregnancy

## Abstract

This study validated the International Physical Activity Questionnaire (IPAQ) and the Context-specific Sedentary Behavior Questionnaire (CSBQ) against accelerometry among parents-to-be. Sex-differences in potential misreporting of physical activity (PA) and sedentary behavior (SB) were also investigated. Self-reported total PA (TPA), light-intensity PA (LPA), moderate-intensity PA (MPA), vigorous-intensity PA (VPA), moderate-to-vigorous-intensity PA (MVPA), and SB of 91 parents-to-be (41 men and 50 women) were compared with Actigraph data according to sex. Furthermore, the extent of misreporting was compared between sexes. Strong correlations for TPA and weak-to-moderate correlations for LPA, MPA, VPA, MVPA, and SB were observed. Participants underestimated TPA by 1068 min/week (=17.8 h/week; −50%), LPA by 1593 min/week (=26.6 h/week; −83%), and SB by 428 min/week (=7.1 h/week; −11%) and overestimated MPA by 384 min/week (=6.4 h/week; +176%) and MVPA by 525 min/week (=8.8 h/week; +224%). Males overreported VPA more than females in absolute minutes per week (238 min/week, i.e., 4.0 h/week vs. 62 min/week, i.e., 1.0 h/week), whereas, in relative terms, the opposite (+850% vs. +1033%) was true. The IPAQ and CSBQ can be used with caution to estimate TPA and SB among parents-to-be considering a strong correlation but low agreement for TPA and a weak-to-moderate correlation but acceptable agreement for SB. We disadvise using these self-reports to estimate PA on the distinct intensity levels.

## 1. Introduction

Life-changing events such as going to college, getting a first job, living together with a partner, and becoming a parent may have an important impact on people’s daily routines [1]. The transition to parenthood, in particular, has been identified as a crucial life-changing event. During this period, parents-to-be may experience several behavioral changes, such as changes in physical activity (PA) and sedentary behavior (SB) [1,2].

In general, insufficient PA levels and excessive SB are associated with a higher risk for developing chronic diseases, such as overweight and obesity, type 2 diabetes, cardiovascular disease, depression, and several types of cancer [3,4,5,6,7]. On the contrary, prenatal PA (performed by the mother) has various health benefits for both the mother and her offspring [8,9,10,11]. Adequate prenatal PA decreases the risk of developing excessive gestational weight gain, gestational diabetes, and prenatal depression and decreases the risk of instrumental delivery and developing macrosomia and other neonatal complications [8,9,10,11]. Besides, higher levels of SB are associated with poorer maternal metabolic outcomes, such as higher levels of C-reactive protein and low-density lipoprotein (LDL) cholesterol, as well as infant outcomes, such as larger newborn abdominal circumference and increased chances of developing macrosomia [12].

PA and SB can be assessed using objective and subjective measurement tools, with both having their own strengths and weaknesses [13]. Objective tools, including accelerometers, provide more accurate and reasonably precise information on the frequency, intensity, and duration of the level of (in)activity, which makes them useful for measuring both PA and SB [14]. One of the drawbacks of these objective methods, however, is the relatively high device cost, together with time and effort constraints, which make them less suitable for large population studies. Furthermore, accelerometers are not able to catalogue the type of activity being performed and are limited in correctly registering activities such as cycling and upper-body and resistance exercises [15,16,17]. Accelerometers are often not water-resistant and should be removed during all water-based activities, which implies that accelerometers are unable to capture aquatic activities [14]. Subjective methods, such as self-report questionnaires, activity diaries, and activity logs, are relatively easy to use and have a low cost compared to objectively measured physical (in)activity. However, these subjective measurement tools are susceptible to recall and social desirability bias [17] and generally lack accuracy in estimating light-intensity PA (LPA), moderate-intensity PA (MPA), and total PA (TPA) [18,19].

Since subjective tools are frequently used to assess PA and SB during pregnancy [12,19,20], it is of interest to investigate the applicability of self-report measurement tools to capture these lifestyle behaviors during this transitional life period. The International Physical Activity Questionnaire (IPAQ) [21] is a widespread and frequently used questionnaire designed to estimate PA levels. The questionnaire takes four different PA activity domains into account, including work-related PA, transport-related PA, domestic and garden PA, and leisure time PA. Besides the assessment of PA levels at different intensities, the IPAQ also includes a question to assess (non-contextual) sitting time. The validity of the IPAQ has already been investigated and established in different populations [21,22]. Two studies have specifically investigated the validity of IPAQ in a population of pregnant women [19,23]. However, these studies investigated PA levels during the second trimester while comparing different measurement tools: IPAQ (long form) vs. Actigraph and IPAQ (short form) vs. SenseWear Armband. Furthermore, both studies reported different correlation coefficients for MPA when comparing the self-report with the objective measurement tool (ρ = 0.09 vs. ρ = 0.38, respectively) [19,23].

To our knowledge, there is no consensus on the most adequate self-report method for assessing SB among parents-to-be in the current literature [12]. As SB typically manifests itself in different contexts (e.g., leisure time, at work, passive transport) and different types (e.g., watching television, computer use, reading) [24,25], it is important to use sufficiently sensitive questionnaires capturing SB levels among those different contexts and types, making the IPAQ less suitable for the assessment of SB levels. The context-specific sedentary behavior questionnaire developed by Busschaert and colleagues [25], herein referred to as the CSBQ, can be used to assess SB in three different contexts, including time spent sedentary during leisure time; time spent sedentary during work, studying, and volunteer work; and time spent sedentary during transport. Busschaert and colleagues [25] found acceptable validity for three different age groups (i.e., adolescents, adults, older adults) when comparing the CSBQ against the activPAL accelerometer. However, the applicability of the CSBQ is yet to be examined in a population of parents-to-be.

Literature on PA and SB during pregnancy has most often focused on mothers-to-be while leaving fathers-to-be aside [26]. A limited number of studies, however, have suggested that pregnancy may equally influence PA and SB levels of fathers-to-be [20,27]. The inclusion of men (besides their pregnant partner) in PA and SB research calls for the validity and agreement testing of self-report PA and SB measures as sex differences may occur. In fact, studies examining over- and underreporting in other populations (i.e., college students) point in the direction of males overreporting vigorous-intensity PA (VPA) more compared to their female counterparts [28,29]. As, to date, no such comparative studies are available in women and men transitioning to parenthood, investigating possible sex differences in misreporting of PA and SB could provide more comprehensive insights in the applicability of self-report questionnaires in this population across sex. Accordingly, the primary objective of the present study was to validate the IPAQ (long form—Dutch version) and an adapted version of the CSBQ against accelerometry (Actigraph) in a population of Belgian pregnant women and fathers-to-be. The secondary objective was to compare potential misreporting of PA and SB levels between sexes.

## 2. Materials and Methods

### 2.1. Design and Participants

Baseline data (around 12 weeks of pregnancy) from the TRANSPARENTS study [30], which is a multicenter observational follow-up study, were used for the present cross-sectional validation study. The overarching protocol has been described in depth elsewhere [30]. In brief, the TRANSPARENTS study is centered around monitoring changes in body weight, body composition, and energy balance-related behavior in both women and men during the transition to parenthood. Between June and December 2018, 152 couples expecting their first child were recruited by means of convenience sampling at the obstetrics units of 4 different hospitals located in Flanders and the Brussels Capital Region (Belgium). Of these 304 Dutch-speaking participants, only women and men for which the questionnaires and accelerometry covered exactly the same time period (i.e., the same 7 days of monitoring the participants’ activities) were used for the present analysis. Participants who did not wear the Actigraph for 7 consecutive days or did not directly complete the questionnaires at the end of the 7-day Actigraph-wearing period were excluded. This resulted in a sample of 95 eligible participants (51 women and 44 men, including 27 couples). This study was conducted in compliance with the principles of the Declaration of Helsinki (current version) and the principles of good clinical practice (GCP) and in accordance with all applicable regulatory requirements. The protocol was approved by the leading Medical Ethics Committee of the University Hospital of the Vrije Universiteit Brussel (UZ Brussel, Brussels, Belgium) on 16 May 2018 (B.U.N. 143201835875). All participants signed an informed consent prior to study participation [31].

### 2.2. Procedure and Measurements

An online questionnaire was used to assess sociodemographic characteristics and subjective PA and SB. Besides, objective measurements were conducted to assess body mass index (BMI), as well as PA and SB. More detailed information on the procedure and measurements can be found in the study protocol [30].

#### 2.2.1. Socio-Demographics

Participants’ sex, age, ethnicity, education, and net family monthly income were assessed through an online questionnaire.

#### 2.2.2. Body Mass Index (BMI)

BMI was calculated by dividing body weight (kg) by squared height (m^2^). Body weight (0.1 kg accurate scale) and height (1 mm accurate scale) were objectively measured using a TANITA MC780SMA Bio-electrical Impedance Analyzer (BIA) and a calibrated SECA wall-mounted stadiometer, respectively.

#### 2.2.3. International Physical Activity Questionnaire (IPAQ)

The long form of the IPAQ (Dutch version) [21,31] was used to estimate TPA levels and PA of different intensities (i.e., LPA, MPA, and VPA). The questionnaire includes 31 items which are distributed over 4 domains (i.e., work-related PA, transport-related PA, domestic and garden PA, and leisure time PA). Within each domain, the participants are required to recall the time spent in different PA intensity levels over the last 7 days, e.g., the amount of LPA (i.e., walking); MPA, including activities such as carrying light loads, cleaning windows, raking in the garden, playing doubles tennis, cycling, or swimming at a regular pace; and VPA, including activities such as heavy lifting, aerobics, running, fast cycling, or swimming. More specifically, participants needed to recall the number of days a certain activity was performed after which the amount of time (hours and minutes) spent on each activity was questioned. Manual data cleaning for outliers and unrealistic data conforming to the IPAQ protocol [32] was conducted on each of the 31 collected items of the questionnaire. As a next step, all responses given in hours were converted into minutes. Subsequently, scores for LPA, MPA, and VPA were calculated within each PA domain separately using the following formula: Min/day spent in activity * days per week. Then, all scores of each domain were combined into a total score (min/week) for LPA, MPA, and VPA. A TPA score was then calculated by summing LPA, MPA, and VPA. A moderate-to-vigorous-intensity PA (MVPA) score was calculated by summing MPA and VPA. Assuming participants spent at least 8 hours per day sleeping, cases scoring unrealistically high values for TPA (i.e., higher than 6720 min/week, which corresponds to 7 days × 960 min/day (or 16 h/day)) or scoring 0 min/day for TPA were excluded from the analysis [32]. Due to the typically skewed nature of the subjective PA data and given the fact that PA levels are typically overreported, a truncation was advised in an attempt to normalize the levels of activity [32]. Therefore, all LPA, MPA, and VPA values above 1260 min/week were truncated (i.e., recoded into 1260 min/week) following the IPAQ protocol [32].

#### 2.2.4. Context-Specific Sedentary Behavior Questionnaire (CSBQ)

An adapted version of the adult version of the CSBQ developed by Busschaert and colleagues [25] was used to estimate SB (in min/week). More specifically, only test items that questioned the amount of sedentary time spent in different contexts across 3 domains (i.e., during leisure time; during work, study and volunteer work; and during transport) were used. Each domain consisted of 1 or more items or activities, such as watching television, making phone calls, eating a meal, and transport by bus, all of which are performed seated or lying down. Test items from the original CSBQ [25] that concerned potential correlates of context-specific SB (e.g., the impact of other family members’ SB), sedentary-related equipment (e.g., the number of televisions and/or computers inside the bedroom), and simultaneous behavior variables (e.g., reading while listening to the radio) were excluded as these items did not measure SB as such. In accordance with the IPAQ, the participants needed to recall the amount of time spent sitting or lying down during the last 7 days within each of the 3 domains. To estimate the number of minutes spent sedentary on those reported days, each test item was filled out using specific time intervals, e.g., 1 to 15 min, 15 to 30 min, 30 to 60 min, 1 h to 2 h, 2 h to 3 h, etc. To calculate the reported number of minutes of each test item, midpoint values of the timeslots (e.g., 7.5 min, 22.5 min, 45 min, etc.) were calculated. Midpoint values of those timeslots that were incalculable (e.g., “more than 7 hours a day” and “more than 8 hours a day” were considered as 450 min and 510 min, respectively). Subsequently, these midpoint values were multiplied by the reported number of days. The total self-reported SB (min/week) was then calculated by adding all values of each test item. In contrast to what was proposed by Busschaert and colleagues [25], no distinction was made between weekdays and weekend days in the present study. For practical and comprehensibility reasons while taking into account the burden on the participants, we opted to align the above with the IPAQ scoring [30,32]. By analogy of the IPAQ data processing [32], all participants scoring SB values above 6720 min/week or 0 min/week were excluded from the analysis.

#### 2.2.5. Accelerometry

Triaxial accelerometers (Actigraph GT3X) were used to objectively measure participants’ PA and SB levels. The participants were instructed to wear the Actigraphs on their right hip for a period of 7 consecutive days. The participants were asked to take off the Actigraph during aquatic activities such as swimming and showering, as well as during night rests [17,33]. The time periods during which participants removed the Actigraph were written down by the participants in an activity log. The collected raw data were processed using ActiLife software (version 6.11.9) [34]. Activity counts, registered by the Actigraph, were summed in epochs of 60 s in order to calculate the total counts per day. Activity bouts during which the Actigraph registered 0 activity counts per 60 consecutive minutes [35] and/or 0 activity counts for a total of 600 (nonconsecutive) minutes per day were excluded from the dataset [36]. TPA, LPA, MPA, VPA, and SB (all measured in min/week) were calculated as outcome measures. For these measurement outcomes, total counts per day were subsequently categorized into 3 PA intensity levels, namely LPA (≥1951 counts/min), MPA (1952–5724 counts/min), and VPA (≥5725 counts/min), conforming to the protocol of Freedson and colleagues [37]. TPA was calculated by summing the LPA, MPA, and VPA values. MVPA was calculated by summing the values of MPA and VPA. For participants who reported to have taken off their Actigraph during aquatic activities, information on the type and intensity of these activities was personally questioned upon return of the activity log. Subsequently, the intensity of the specific aquatic activity was quantified to conform to the classifications of Ainsworth and colleagues [38], and the activity was categorized as LPA, MPA, or VPA. These values were manually added (according to the associated estimated intensity) to the final Actigraph PA measurement outcomes.

### 2.3. Data Analysis

All data were analyzed using IBM SPSS Statistics version 27 (IBM, Armonk, NY, USA). Q-Q plots and Kolmogorov–Smirnov tests were used to test normality of the data distributions. Since the IPAQ outcomes (i.e., estimated means of TPA, LPA, MPA, VPA, and MVPA, all in min/week) were not normally distributed, Spearman rank correlation coefficients (ρ) were calculated to assess concurrent validity between the PA outcomes measured by the IPAQ and the Actigraph. In analogy, Pearson correlation coefficients (r) were calculated with the normally distributed data (i.e., SB outcomes). Wilcoxon signed rank tests (IPAQ vs. Actigraph outcomes) and paired samples *t*-tests (CSBQ vs. Actigraph outcomes) were used to assess agreement in subjectively and objectively measured PA and SB, respectively. For each outcome (estimated means of TPA, LPA, MPA, VPA, MVPA, and SB), a difference score was calculated by subtracting the questionnaire mean value (in min/week) with the Actigraph mean value (in min/week). The difference score was then divided by 60 and expressed in hours/week. To determine the relative extent of under- or overestimation for each outcome, the difference score (in min/week) was divided by the Actigraph mean value (in min/week) and multiplied by 100. The difference score is thus expressed as a percentage of deviation, with negative values representing underestimations and positive values representing overestimations. Lastly, to be able to compare the relative extent of misreporting between the different outcomes, a misreporting ratio (MR) was calculated by dividing the highest mean value by the lowest mean value. For the above calculations, mean values were used as, in some cases, medians were “zero,” making it impossible to calculate the aforementioned ratio. Therefore, means and standard deviations (SD) as well as medians and interquartile ranges (IQR), are reported. Finally, to compare difference scores between males and females, Mann–Whitney U tests were executed for TPA, LPA, MPA, VPA, and MVPA (i.e., not normally distributed data), while independent samples *t*-tests were applied for SB (i.e., normally distributed data). *p*-values < 0.05 were considered statistically significant.

## 3. Results

### 3.1. Sample Characteristics

The sociodemographic and anthropometric characteristics of the participants are displayed in Table 1. Of the initial 95 participants, 2 men were excluded because of scoring 0 min/week on TPA, 1 man because of scoring more than 6720 min/week on TPA, and 1 woman because of scoring more than 6720 min/week on total SB, resulting in a final sample of *n* = 91 (41 men and 50 women, including 26 couples). In total, 7 IPAQ observations were truncated for LPA, 22 for MPA, and 3 for VPA. Six participants’ Actigraph data were manually adjusted for aquatic activities. Mean Actigraph wear time was 14.7 ± 1.0 h/day (range = 12.3–16.8 h/day).

### 3.2. Validity of the IPAQ and CSBQ Data Compared to the Actigraph Data

Spearman rank correlations (ρ) between the IPAQ and the Actigraph for TPA and each PA intensity level (LPA, MPA, VPA, and MVPA), as well as Pearson correlations (r) between CSBQ and the Actigraph for SB, are shown in Table 2. All correlation coefficients were found to be positive. For the total sample, a significant and strong correlation was found for TPA (ρ = 0.664), while significant but weak to moderate correlations were found for LPA (ρ = 0.243), VPA (ρ = 0.353), and MVPA (ρ = 0.323). Male-only data showed a significant and strong correlation for TPA (ρ = 0.614) but a significant and weak-to-moderate correlation for MVPA (ρ = 0.342). Female-only data showed strong significant correlations for TPA (ρ = 0.559) and VPA (ρ = 0.506). No significant correlations were found for MPA. Lastly, a significant but weak-to-moderate correlation coefficient was found for SB in the total sample (r = 0.340), as well as in fathers-to-be (r = 0.388).

### 3.3. Agreement between Subjectively and Objectively Measured PA and SB

Means and SDs and medians and IQRs of the different PA and SB levels, combined with Z- (PA) or t- (SB) values, are also presented in Table 2. Significant differences in estimates of min/week between the questionnaires and the Actigraph were found for all variables and both sexes. On average, participants underestimated their TPA by 1068 min/week (i.e., 17.8 h/week; −50%; MR = 2.0). LPA was underreported, on average, by 1593 min/week (i.e., 26.6 h/week; −83%; MR = 5.9). For MPA, participants overestimated their values by 384 min/week (i.e., 6.4 h/week; +176%; MR = 2.8) on average. No sex differences in misreporting were observed for TPA (*p* = 0.908), LPA (*p* = 0.175), or MPA (*p* = 0.820). For VPA, males misreported more than females in absolute minutes per week (*p* = 0.009). On average, fathers-to-be overreported their VPA by 238 min/week (i.e., 4.0 h/week; +850%; MR = 9.5) and pregnant women by 62 min/week (i.e., 1.0 h/week; +1033%; MR = 11.3). Participants overreported their MVPA by an average of 525 min/week (i.e., 8.8 h/week; +224%; MR = 3.2). Finally, participants underreported their SB by 428 min/week (i.e., 7.1 h/week; −11%; MR = 1.1) on average. No sex differences in misreporting were observed for MVPA (*p* = 0.063) and SB (*p* = 0.964).

## 4. Discussion

Compared to accelerometry, the IPAQ shows good validity for TPA (ρ = 0.664), with underestimations of around 50%. Weak-to-moderate validity was observed for the distinct PA intensity levels (ρ-values ranging from 0.057 to 0.506) with poor agreement (misreporting ranging from −83% to +1033%). The CSBQ showed weak-to-moderate validity (r-values ranging from 0.277 to 0.388) with acceptable agreement (underreporting by 11%). Except for VPA, no sex differences in misreporting were observed.

The most important finding is that despite of a relatively good consistency (i.e., strong correlation), pregnant women and fathers-to-be underestimated their TPA by 17.8 h/week (which corresponds to −50%). Underreporting of TPA in pregnant women was also observed by Harrison and colleagues [19]. In contrast and regardless of sex, MVPA was overestimated by 8.8 h/week in the present study (which corresponds to +224%), with low consistency between the IPAQ and the Actigraph. The overreporting of MVPA is also consistent with other research [39]. In fact, Brett and colleagues observed an even greater degree of overreporting in MVPA of 12.1 h/week in pregnant women [39] compared to 7.1 h per week for females in the present study. It should be mentioned that the study conducted by Brett and colleagues included women at 13–28 weeks of gestation, did not include women’s partners and used another PA measurement tool, namely the Pregnancy Physical Activity Questionnaire (PPAQ) [39,40]. In contrast, another study by Watson and colleagues found no significant difference in MVPA between accelerometry and the Global Physical Activity Questionnaire (GPAQ) at 14–18 weeks of gestation [41]. It is particularly important to accurately estimate MVPA, as PA recommendations are primarily based on MVPA levels since its health benefits are well established [42].

The overall underestimation of TPA in the present study was caused by an overproportionate underestimation of LPA of 26.6 h/week (which corresponds to −83%). Moreover, LPA showed low consistency between the IPAQ and the Actigraph, which is again in accordance with the study by Harrison and colleagues [19]. The relative underreporting of LPA (expressed in terms of MR = 5.9) was almost twice as high as the overreporting of MVPA (MR = 3.2). It should be said that the IPAQ only considers time spent walking to estimate LPA, which means that low-intensity activities other than walking are missed. LPA is defined as “PA that is performed between 1.5 and 3 METs (Metabolic Equivalent of Task). On a scale relative to an individual’s personal capacity, LPA is usually a 2–4 on a rating of perceived exertion scale of 0–10” [42]. This includes all activities that do not result in a substantial increase in heart rate or breathing rate [42]. This may (partially) explain the quite large underestimation that we observed compared to the Actigraph. Another possible explanation for this striking finding may be that moderate and vigorous-intensity activities are easier to remember than light-intensity activities due to their association with the feeling of exhaustion [43]. The accuracy of measurement of LPA might be even more important than generally thought. A recent systematic review concluded that objectively measured LPA is beneficially associated with important health outcomes (i.e., lower all-cause mortality and decreased cardiometabolic risk factors such as triglyceride levels and metabolic syndrome) after adjustment for MVPA in the adult population [44]. This is important as, especially during the transition to parenthood, LPA becomes more prominent compared to MVPA in both sexes [20,27].

Interestingly, we found that fathers-to-be overreported VPA more than pregnant women in absolute minutes per week (+238 vs. +62 min/week), which is in agreement with the findings of Dyrstad and colleagues [43]. However, when considering overreporting in relative terms (i.e., percentage of deviation), we observed the opposite (+850 vs. +1033%), which is not in accordance with the above-referenced study [43]. Besides the difference in questionnaire methodology (i.e., comparing the IPAQ short form with the Actigraph) and study population (i.e., using a general population sample of 1751 adults aged 19–84 years old) [44] compared to the present study, this discrepancy may be explained by alterations in perceived intensity of PA by pregnant women. Due to general fatigue or other discomforts related to pregnancy, activity intensities may be experienced as more strenuous than before [45]. Pregnant women may therefore misreport VPA to a greater extent when compared to fathers-to-be, who do not experience extra fatigue or physical discomforts. In fact, one could argue whether or not this should be classified as a ‘misreport.’ VPA is defined as “PA that is performed at 6 or more METs. On a scale relative to an individual’s personal capacity, VPA is usually a 7 or 8 on a rating of perceived exertion scale of 0–10” [42]. This means that vigorous activities result in much stronger breathing, more sweating, and notably increased heart rates [46]. However, due to alterations in the cardiovascular system (both anatomically and physiologically) of pregnant women leading to an increased heart rate during the first trimester of pregnancy [47], the ‘load’ of certain activities on the cardiovascular system may indeed not only be perceived as more vigorous but is actually more vigorous from a physiological point of view. The question then arises whether the accelerometer is still the preferred measurement tool to measure VPA in pregnant women. Although other objective tools measuring physiological parameters such as heart rate (e.g., via heart rate monitors) or skin temperature (e.g., via SenseWear) may be (part of) the solution, they still have their own limitations to take into account [48,49]. It might therefore be advisable to use objective and subjective measures of PA in tandem. For example, accelerometry can be used alongside activity logs where people have to rate their perceived exertion (e.g., via the Borg ratings of perceived exertion scale [50]). Moreover, both accelerometry and PA logging and rating may be integrated in smartphones or smartwatches, making it easier to measure daily activities on a larger scale and over a longer time period. Smartphones are increasingly integrated in scientific research and have been used in the past to measure PA [51]. Furthermore, a recent study demonstrated the use and value of smartphones in facilitating behavioral interventions (including PA) in pregnant women [52].

For SB, weak-to-moderate correlations were observed along with a self-report underestimation by 7.1 h/week (which corresponds to −11%) relative to the Actigraph. Oviedo-Caro and colleagues [53] similarly observed a weak correlation and agreement between the Sedentary Behavior Questionnaire (SBQ) [54] and the SenseWear Armband. It should be mentioned, however, that the latter study was performed in the third trimester of gestation. Hence, self-reports of SB may be influenced by an increased tiredness or other degrees of discomfort such as nausea, heartburn, and musculoskeletal complaints compared to the first trimester of pregnancy, which was used as time window in the present study. No comparable studies were found for fathers-to-be, highlighting the uniqueness of the present study.

Our findings, along with others [19,39,55], show that both PA and SB self-reports during pregnancy (for both women and men) are prone to considerable bias. Recall and social desirability bias are typical and well-known biases inherent to self-reports, causing these (un)systematic errors [17]. In addition, it seems likely that people struggle with correctly classifying different intensities (e.g., light vs. moderate PA or moderate vs. vigorous PA), possibly resulting in a misclassification bias upon self-report. In fact, the latter can be demonstrated by the relatively strong correlations found for TPA, while only weak-to-moderate correlations were observed when splitting it up in LPA, MPA, and VPA. This indeed suggests misclassification errors across intensities while still showing good overall (TPA) validity. As discussed above, misclassification bias may be equally present with accelerometry in pregnant women. Moreover, studies reporting on PA and SB self-reports differ in both methodology (i.e., measurement tools, data cleaning, data processing, cut-off points for truncation or exclusion) and statistical analysis [35,36,56]. A systematic review including a total of 148 studies showed a range of under- and overreporting in TPA of −78% to +500%, with an average overreport of 44% in the general adult population (males and females combined) [57]. Interestingly, the same review established higher overreporting by females (138%; range = −100% to 4024%) compared to males (44%; range = −100% to 425%). It is clear that both interpretation and comparison of PA and SB self-reports should be performed with extreme caution.

One could argue whether the observed under- and overestimations in PA and SB are fully due to methodological inaccuracies and not to, for example, a lack of understanding of the purpose and procedures of the study by the participants. As mentioned, the present study is part of the larger TRANSPARENTS study [30] in which participants were recruited face-to-face at the participating hospitals. During the recruitment phase, couples who were expecting their first child were carefully informed about the nature, purpose, and course of the study. During the data collection phase, the principal investigator visited all participating couples at their respective homes in order to perform objective measurements, such as body composition measures (see study protocol [30]), but also to give instructions concerning wearing the Actigraph correctly. Participants were carefully explained when and how they should wear the device and fill out the accompanying activity log as well as when and how they should fill out the IPAQ and CSBQ. One week after each home visit, an email was sent to remind the participating couple of the fact that they should fill out the IPAQ and CSBQ (i.e., immediately after 7 days of wearing the Actigraph). Furthermore, a website (https://www.transparents.be (accessed on 23 April 2021)) was developed for the participants in order to (re)inform them about the nature of the study. Due to the close contact approach within the study, it seems unlikely that the participants were not well informed about the nature, purpose, and procedures of the study.

Both researchers and health care providers should be aware of the extent of misreporting as well as the fact that both under- or overreporting may be present depending on the (perception of) energy expenditure behavior and/or intensity level. In this respect, the erroneous measurement and its implications have been described by Guérin and colleagues [55]. In their commentary paper, the authors warn that conclusions about health outcomes based on unreliable PA estimates could lead to misinformed clinical recommendations and sway future research [55]. Of course, the same is true for SB. Although self-report measures may be important as they provide information on the purpose, type, and context of PA and SB [58], objective measures seem absolutely necessary (preferably in combination with subjective reporting of activity type and intensity) to pursue valid and reliable PA and SB estimates in parents-to-be.

A first strength of the present study is the inclusion of fathers-to-be alongside the pregnant women. Therefore, it was possible to describe validity (in)differences between males and females during the transition to parenthood. Second, the use of activity logs during the Actigraph wear time period enabled us to do manual data cleaning and imputation (e.g., aquatic activities which otherwise would have been addressed as “non-wear time”). The latter approach increased the accuracy of the Actigraph output. A third strength is the breakdown of TPA into LPA, MPA, VPA, and MVPA while also including SB, providing useful validity insights per energy expenditure behavior.

A first limitation of the present study is that our sample consisted of pregnant women and fathers-to-be who volunteered to participate, which resulted in self-selection bias. For example, our sample predominantly had high levels of education. Although considerable overreporting was detected (e.g., for MVPA), more highly educated people are generally expected to estimate their PA and SB levels more accurately [59]. So, misreporting may be even greater in a more representative sample of parents-to-be. Second, the present study validated the IPAQ, while other self-report tools are available to specifically assess PA in a population of pregnant women [60]. As explained above, we used data from the ongoing TRANSPARENTS study [30], which focuses on both women and their partner during pregnancy and the postpartum period. Third, on 26 November, the World Health Organization (WHO) launched its new guidelines on PA and SB [42]. One of the biggest changes compared to the previous guidelines is that “every move counts” [42]. As the IPAQ was developed according to the previous WHO PA guidelines, it only takes minimal activity bouts of 10 min into account. This also means that bouts less than 10 min are converted to 0 min when following the IPAQ protocol [32]. This obviously results in PA being underestimated. Newly developed or at least updated PA measurement tools and scoring protocols are needed to better align with the new WHO PA guidelines. These self-report tools should be clear and understandable, with a clear description of the different PA intensities and sufficiently padded with obvious and everyday examples, in order to minimize misclassification bias. As pregnant women may perceive physical activities as more demanding compared to nonpregnant women, it might be advisable to let people rate their own perceived exertion, leaving the classification as such to the assessor. Furthermore, the scoring protocols should be clear and sufficiently detailed, leaving no room for misinterpretations upon data processing and analysis. Fourth, although the observed Actigraph wear time was satisfactory (mean = 14.7 ± 1.0 h/day; range = 12.3–16.8 h/day), the non-wear time may have biased our results. Assuming people usually sleep around 8 h/night, non-wear time for some participants may have been as high as 4 h/day. Because the non-wear time periods were mainly present during the evening (when participants took off the Actigraph device), it is most likely that particularly SB (e.g., watching television from the couch before going to sleep) was underestimated by the Actigraph. The latter suggests that the observed agreement between the CSBQ and Actigraph may be (slightly) overestimated. Fifth, due to the nonnormal distribution of the IPAQ data, nonparametric tests were performed, while Intraclass Correlation Coefficients (ICC) or Bland–Altman tests (i.e., parametric validity tests for which normality is an important assumption) might have provided more accurate results. However, due to the typical skewness of the data distribution delivered by the IPAQ, the IPAQ protocol specifically recommends using nonparametric testing [32]. Finally, because, in some cases, the observed medians were “zero,” we calculated the difference scores, percentages of deviation, and MRs based on the mean values of the IPAQ and Actigraphs. Given the abovementioned nonnormality issue of PA outcomes, these calculated values should be interpreted with caution. Nonetheless, these values should give an idea on the extent to which self-reports under- or overestimate objectively measured PA and SB. For the same reasons, mean and SD as well as median and IQR are reported.

## 5. Conclusions

The IPAQ showed good validity in the total sample for TPA (ρ = 0.664) compared to the Actigraph data. Accordingly, the IPAQ can be used to assess TPA in a population of parents-to-be, though underestimations around 50% should be taken into account. The IPAQ showed weak-to-moderate validity for LPA, MPA, VPA, and MVPA (with ρ-values ranging from 0.057 to 0.506) and large under- and overestimations (ranging from −83% to +1033%). Males (+238 min/week) overreported VPA more than females (+62 min/week) in absolute minutes per week whereas, in relative terms, the opposite was found to be true (+850% vs. +1033%, respectively), which may be due to alterations in perceived intensity when being pregnant. The CSBQ showed weak-to-moderate validity (r-values ranging from 0.277 to 0.388) and underreporting by 11%. So, some caution should be taken when using both the IPAQ and the CSBQ to estimate PA and SB in parents-to-be. Therefore, we recommend using objective measures, such as accelerometry, to accurately estimate PA and SB. Objective measures should be accompanied by subjective measures, such as activity logs, to create a more thorough understanding of the context, type, and perceived intensity of the performed activities.

## Figures and Tables

**Table 1 ijerph-18-04654-t001:** Sample characteristics (mean ± SD, %).

Outcome Measures	Total (*n* = 91)	Male (*n* = 41)	Female (*n* = 50)
Age (years)	29.7 ± 3.3	30.6 ± 3.1	28.9 ± 3.3
BMI (kg/m^2^)	24.6 ± 4.1	25.1 ± 3.5	24.2 ± 4.6
Weeks of gestation (of the partner) at study entrance	12.9 ± 1.0	12.9 ± 1.1	12.9 ± 1.0
Ethnicity (%)			
Caucasian	98.9	97.5	100.0
Other	1.1	2.5	0.0
Education (%)			
Primary education	6.8	12.5	2.1
Secondary education	15.9	25.0	8.3
Higher education	77.3	62.5	89.6
Net family monthly income (%)			
Less than €3000	6.6	7.3	6.0
Between €3000 and €4000	50.5	51.2	50.0
Between €4000 and €5000	30.8	31.7	30.0
More than €5000	12.1	9.8	14.0

SD = standard deviation; BMI = body mass index (objectively measured).

**Table 2 ijerph-18-04654-t002:** Validity and agreement of IPAQ and CSBQ data vs. Actigraph data.

Outcome Measures	Questionnaire (IPAQ/CSBQ)		Actigraph		Spearman ρ (PA) or Pearson r (SB)	Z-Value (PA) or t-Value (SB)
	Mean ± SD	Median (IQR)	Mean ± SD	Median (IQR)		
TPA (min/week)						
All (*N* = 91)	1087 ± 814	810 (1150)	2155 ± 548	2088 (663)	0.664 ***	−8.054 ***
Males (*n* = 41)	1341 ± 875	1160 (1243)	2367 ± 566	2279 (858)	0.614 ***	−5.242 ***
Females (*n* = 50)	878 ± 703	648 (1080)	1980 ± 470	1935 (721)	0.559 ***	−6.096 ***
LPA (min/week)						
All (*N* = 91)	328 ± 362	180 (460)	1921 ± 491	1834 (688)	0.243 *	−8.284 ***
Males (*n* = 41)	395 ± 398	240 (548)	2071 ± 503	2045 (820)	0.154	−5.579 ***
Females (*n* = 50)	273 ± 324	175 (336)	1797 ± 450	1722 (731)	0.232	−6.154 ***
MPA (min/week)						
All (*N* = 91)	602 ± 463	450 (1000)	218 ± 124	196 (155)	0.184	−6.174 ***
Males (*n* = 41)	681 ± 476	525 (1005)	269 ± 130	251 (205)	0.229	−4.192 ***
Females (*n* = 50)	538 ± 447	380 (825)	176 ± 102	151 (134)	0.057	−4.511 ***
VPA (min/week)						
All (*N* = 91)	157 ± 282	20 (180)	16 ± 40	0 (13)	0.353 ***	−5.329 ***
Males (*n* = 41)	266 ± 334	135 (435)	28 ± 55	1 (35)	0.097	−4.242 ***
Females (*n* = 50)	68 ± 192	0 (60)	6 ± 15	0 (2)	0.506 ***	−3.440 ***
MVPA (min/week)						
All (*N* = 91)	759 ± 616	500 (975)	234 ± 142	207 (166)	0.323 **	−6.963 ***
Males (*n* = 41)	947 ± 657	810 (1155)	297 ± 154	263 (243)	0.342 *	−5.008 ***
Females (*n* = 50)	606 ± 539	393 (833)	183 ± 109	151 (150)	0.136	−4.775 ***
SB (min/week)						
All (*N* = 91)	3591 ± 1120	3562 (1711)	4019 ± 584	4014 (879)	0.340 ***	−3.801 ***
Males (*n* = 41)	3466 ± 1095	3389 (1879)	3899 ± 676	3940 (1231)	0.388 *	−2.666 *
Females (*n* = 50)	3694 ± 1141	3629 (1732)	4117 ± 481	4140 (706)	0.277	−2.696 **

* *p* < 0.05; ** *p* ≤ 0.01; *** *p* ≤ 0.001; IPAQ = International Physical Activity Questionnaire; CSBQ = Context-specific Sedentary Behavior Questionnaire; IQR = interquartile range; SD = standard deviation; TPA = total physical activity; LPA = light-intensity physical activity; MPA = moderate-intensity physical activity; VPA = vigorous-intensity physical activity; MVPA = moderate-to-vigorous-intensity physical activity; SB = sedentary behavior.

## Data Availability

The datasets used and/or analyzed during the current study are available from the corresponding author on reasonable request.

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
