# Peer review of "Misreporting of Physical Activity and Sedentary Behavior in Parents-to-Be: A Validation Study across Sex"

_ijerph, 2021, doi:10.3390/ijerph18094654_

Round 1
Reviewer 1 Report
Your paper has been sent for my consideration. The manuscript touches on a very interesting and current topic with explanation of an existing knowledge gap. The study attempts to validate the international Physical Activity Questionnaire (IPAQ) and the Context-specific Sedentary Behavior Questionnaire (CSBQ) against accelerometery among parents-to-be. An important element (innovation, improvement) of the work is the incorporation of fathers-to-be in the research group. The advantage of the work is the comparison of the various methods and indication of the limitations with the use of IPAQ and CSBQ.
I only have a few comments and suggestions which are listed below.
[245] Please clarify if the people who overestimated the IPAQ and CSBQ results were included in the target group n = 96 or in the general group n = 304 only?
[306] If there is a representative group in the study, it is necessary to refer to the calculation method used to determine the representative group size.
[310] Were all future parents expecting their first children?
[320] Does the ethnicity specified in the study matter? I believe that including racial origin is unnecessary.
[359] Are the underestimation of the TPA measurement and the overestimation of the MVPA measurement resulted from a faulty method, research group or lack of understanding of the method and purpose of the study by patients? This should be described with more details in the discussion part, including review of available publications.
[463] The text mentions that interpretation and comparison of PA and SB self-reports should be made with extreme caution. Is it possible to propose a better, more accurate method of measuring physical activity and a sedentary lifestyle to avoid such a large deviation?
Perhaps in the future, smartphones/smartwatches could be used as tools for monitoring and recording physical activity and rest time.
I am proposing to use the publication:
- Zuccolo PF, Xavier MO, Matijasevich A, Polanczyk G, Fatori D. A smartphone-assisted brief online cognitive-behavioral intervention for pregnant women with depression: a study protocol of a randomized controlled trial. Trials. 2021;22(1):227. doi:10.1186/s13063-021-05179-8.
Additionally, I would suggest following the testimonials that may be helpful:
- Ferrari GL de M, Kovalskys I, Fisberg M, et al. Comparison of self-report versus accelerometer – measured physical activity and sedentary behaviors and their association with body composition in Latin American countries. McLester CN, ed. PLoS One. 2020;15(4):e0232420. doi:10.1371/journal.pone.0232420
- Sattler MC, Jaunig J, Tösch C, et al. Current Evidence of Measurement Properties of Physical Activity Questionnaires for Older Adults: An Updated Systematic Review. Sport Med. 2020;50(7):1271-1315. doi:10.1007/s40279-020-01268-x
- Felez-Nobrega M, Bort-Roig J, Briones L, et al. Self-reported and activPALTM-monitored physical activity and sedentary behaviour in college students: Not all sitting behaviours are linked to perceived stress and anxiety. J Sports Sci. 2020;38(13):1566-1574. doi:10.1080/02640414.2020.1748359
- Watson ED, Micklesfield LK, van Poppel MNM, Norris SA, Sattler MC, Dietz P. Validity and responsiveness of the Global Physical Activity Questionnaire (GPAQ) in assessing physical activity during pregnancy. Tauler P, ed. PLoS One. 2017;12(5):e0177996. doi:10.1371/journal.pone.0177996
Reviewer 2 Report
The authors wrote in conclusions: "The IPAQ showed good validity in the total sample for TPA (ρ=0.664) compared to the Actigraph data. Accordingly, the IPAQ can be used to assess TPA in a population of
parents-to-be, though underestimations around 50% should be taken into account."
In my opinion, this should be clarified as it is a contradiction. Is IPAQ worth attention and use or not?
Reviewer 3 Report
This is, in summary, an interesting study aimed to validate the IPAQ (long form – Dutch version) and an adapted version of the CSBQ against accelerometry as well as to compare potential misreporting of f physical activity (PA) and sedentary behavior (SB) levels between sexes. The authors found that strong correlations for Self-reported total PA (TPA) and weak to moderate correlations for light-intensity PA (LPA), moder25 ate-intensity PA (MPA), vigorous-intensity PA (VPA), moder26 ate-to-vigorous-intensity PA (MVPA) and SB. Participants underestimated TPA, LPA, SB, and overestimated MPA and MVPA. Males overreported VPA more than females in absolute minutes per week whereas in relative terms the opposite was true.
The authors may find as follows my main comments/suggestions.
First, when throughout the Introduction section, the authors correctly reported that SB is associated with a higher risk for developing chronic diseases, such as cardiovascular diseases and depression, they could even briefly mention the importance of childhood abuse/maltreatment to this regard. Importantly, childhood traumatic experiences need to be considered a potential risk factor for depression, SB and suicidal behavior. The increased vulnerability to depression seems to be, at least partially, related to experiences of childhood traumatic experiences. Thus, given the importance of this topic (although i understand that the link between traumatic experiences, depression and suicidality is not the main topic of the present paper), i suggest to cite within the main text the article published on Frontiers in Psychiatry in 2017 (PMID: 28970807).
Moreover, as the most relevant aims/objectives have been clearly reported within the main text, the main hypotheses should be similarly reported.
Furthermore, the the long form of the IPAQ might be described more succinctly.
Importantly, the authors might immediately present and discuss, in the first lines of the Discussion section, their most relevant study findings. Conversely, they seem to focus on the main aims/objectives that should have been stressed elsewhere within the main text.
Also, the most relevant shortcomings/limitations should be reported in a more detailed manner as the description of the main caveats has been only partially provided.
Importantly, what is the take-home message of this manuscript? While the authors reported that the the IPAQ showed good validity in the total sample for TPA (ρ=0.664) compared to the Actigraph data, here they should provide their conclusive remarks for the readers. Specifically, why caution is recommended when using both the IPAQ and the CSBQ to estimate PA and SB in parents-to-be? What are the most relevant implications of this study? Here, some additional details/information are needed.
Round 2
Reviewer 3 Report
In the revised manuscript, the authors addressed sufficiently most of the major questions raised by Reviewers.